# Essential Design Strength and Unified Strength Condition of ETFE Membrane Material

**DOI:** 10.3390/polym14235166

**Published:** 2022-11-27

**Authors:** Mingyue Zhang, Yingying Zhang, Guangchun Zhou, Hanyin Li

**Affiliations:** 1School of Civil Engineering, Harbin Institute of Technology, Harbin 150090, China; 2Key Lab of Structures Dynamic Behavior and Control, Ministry of Education, School of Civil Engineering, Harbin Institute of Technology, Harbin 150090, China; 3Key Lab of Smart Prevention and Mitigation of Civil Engineering Disasters of the Ministry of Industry and Information Technology, School of Civil Engineering, Harbin Institute of Technology, Harbin 150090, China; 4School of Mechanics and Civil Engineering, China University of Mining and Technology, Xuzhou 221116, China; 5College of Forestry, Henan Agricultural University, Zhengzhou 450002, China

**Keywords:** ETFE membrane material, state-of-stress, characteristic point, essential design strength, unified strength condition

## Abstract

This study proposes essential design strength and unified strength condition for ETFE membrane materials based on the structural state-of-stress theory and formula of strength. Firstly, the tested strain data of the uniaxial rectangle-shaped specimen are modeled to obtain its state-of-stress characteristic parameter. Then, the characteristic points in the evolution curve of the characteristic parameter are detected by the cluster analysis (CA) criterion. The characteristic points are the embodiment of the natural law from quantitative change to qualitative change of a system, which define the essential strength and the essential design strength of ETFE membrane materials. Further, the essential principal stresses are derived at the characteristic points in the evolution curves of the characteristic parameters obtained by the state-of-stress analysis of the strain data from the tests of air bubbling models and cruciform specimens. Both essential principal stresses and essential strength lead to the unified formula of strength for ETFE membrane materials. Additionally, the unified strength condition is derived for the design of ETFE membrane material structures. Finally, the essential strength, essential design strength, and the unified strength conditions are compared with the existing conditions, providing a rationality to update the existing analysis and design methods for determining the strength of ETFE membrane materials.

## 1. Introduction

ETFE materials were first used in construction in 1982 to replace the failed fluorinated ethylene propylene (FEP) film on the roof of Burgers’ Zoo Mangrove Hall in Arnhem, Netherlands [1]. Subsequently, ETFE has been applied to engineering structures, such as the Eden Project in the UK, National Space Center, Zurich Zoo-New Masoala Rainforest in Switzerland, and National Swimming Center in China [2]. Design methods for ETFE material structures were also proposed to further develop these materials. Among them, the safety factor design (SFD) method was commonly adopted in the existing design codes. However, the SFD method was not accurate in some cases because of the uncertainty of ETFE membrane material strength and the empirical/statistical basis. As a result, the different strength indexes of ETFE membrane materials were proposed for design codes [3]. In total, the strength and design of ETFE membrane material have been following the conventional methods, such as the investigations of strength into other engineering materials such as concrete and metal:The material strength was commonly defined referring to the ultimate/peak loads of uniaxial specimens, which had the inherent uncertainty closely related to the initial defects of the material and the shape/size of the specimen [4]. Additionally, the ultimate strength was empirically reduced as the strength index of the material. Consequently, the same material could have different strength indexes in individual design codes and engineering handbooks [5]. The other most common way to define the strength of a material is to stipulate a specific strain value, for instance, the strength of stainless steel was defined by 0.2% of proof stress [6]. For the ETFE membrane material, its strength was defined by a specific strain, 5% of the specimen’s residual strain [7].For the material strength in the combined principal stresses (biaxial or triaxial specimens), there is no strength theory suitable for various materials with different failure forms, so that up to a hundred of strength theories have been proposed to deal with the material strength under complex stress states in the past research history of strength [7,8,9]. In other words, there is no general formula of strength to relate the material strength (uniaxial strength) with the combined principal stresses for various materials. The unified strength theory establishes a unified formula of strength, but it still includes two empirical parameters [10]. The cross biaxial strength of ETFE membrane material was simply estimated as about 45–60% of the uniaxial strength, and there was not a definite relation with the uniaxial strength [11].

Evidently, conventional methods were difficult to achieve the consistent and definite strength for ETFE membrane materials. However, the efforts to overcome the difficulty have been paid by researchers and engineers, which could simply divide three stages: the purely empirical stage, the safety factor stage, and the probability theory stage. The last two stages are reviewed as follows. 

The safety factor stage was based on the in-depth study of the mechanical properties of materials. This stage takes the resistance R and the action effect S to express the structural safety. The safety factor n>1 was introduced into structural design, that is,
(1)$$n=\frac{R}{S}$$

The red dotted lines in Figure 1 show the probability density functions $f_{R}(r)$ and $f_{S}(s)$ of R and S, as plotted using red dotted lines. Since R and S are both single-valued ignoring uncertainty and are determined without clear mechanical mechanisms, n cannot fully and accurately reflect the safety reserve of the structure or its components. 

In the probability theory stage, R or S was as a random variable that obeys a certain distribution pattern for the design of a structure. For two structures with the same mean values ($\mu_{R}$, $\mu_{S}$) of R or S, the safety factors are also the same. However, the structures are not necessarily equally safe or reliable because they also depend on the distribution patterns of R and S. As shown in Figure 1, two distribution functions $f_{R}(r)$ and $f_{S}(s)$ (black solid curves) did not overlap, so the structure would not be damaged. When the discrete degree of R or S increases, $f_{R}(r)$ and $f_{S}(s)$ (black dotted curves) overlap each other, indicating that R played more of a role, and the structural reliability decreased, but *n* did not change [12]. For instance, the current Chinese standard, “Technical Regulations for Membrane Structures” (CECS 158:2015), adopts the following formula:(2)$$K(S_{G_{K}}+S_{Q_{K}})\leq R_{k}$$
where $R_{k}$ is the standard resistance value of the membrane structure; $S_{G_{K}}$ is the standard value of the permanent load effect; $S_{Q_{K}}$ is the standard value of the variable load effect; and K is the safety factor within 3~4 and 6~8 under short-term and long-term loads, respectively. In the EU code, K=3~8.3 depend on the extent of contamination, the quality of membrane surface, the quality of thermal fusion and the membrane area [13]. In the American code, K=3~8 depend on individual loading cases [14,15]. In the Japanese code, K=4 and K=8 are for short-term and long-term loads, respectively [16]. In the Chinese code, K=3.5 and K=7 are for short-term and long-term loads, respectively [6].

Generally, the structural reliability design method based on the probability theory also needed to calculate the partial coefficient in the design of membrane structures [17]. The resistance/partial factor method refers to the specimen’s ultimate strength: (3)$$f=\zeta \frac{f_{K}}{\gamma_{R}}$$
where f is the design value of tensile strength; $f_{K}$ is the standard value of tensile strength; ζ is the strength reduction factor; $\gamma_{R}$ is the partial coefficient of membrane material resistance, which is expressed by Equation (4): (4)$$\gamma_{R}=\frac{K(S_{G_{K}}+S_{Q_{K}})}{\gamma_{G}S_{G_{K}}+\gamma_{Q}S_{Q_{K}}}$$
where $\gamma_{G}$ and $\gamma_{Q}$ are the partial factors for the permanent and variable loads, respectively. Beyond the methods mentioned above, Wu proposed a geometrical method to determine the standard strength of ETFE membrane materials [18]; this method can derive a more rational design strength than previous methods using the coefficients of safety. Additionally, Wu estimated that the cross biaxial strength based on the specimen’s ultimate state is about 45–60% of the uniaxial strength [11]. Overall, the strength of ETFE membrane materials was found to be within a stipulated range and the coefficient of safety was determined not to be an accurate parameter rather than an empirical one. Additionally, the uniaxial strength and biaxial principal stresses of ETFE membrane materials do not have a unified formula to relate them. This situation has led to the overuse of materials to avoid and minimize the negative effects of material strength inaccuracy in design and engineering practice. Indeed, this situation has created a paradigm in the research field of material strength [19], and researchers/engineers know that even the overuse of materials would not warrant affecting the safety of engineering structures in some cases. Hence, reliability was introduced into engineering designs to address the issue of safety due to the inaccuracy of both material strength and structural bearing capacity. 

In sum, past researches on material strength yielded two common conclusions: (a) the strength of a material is certainly related to the specimen’s shape and size, known as the size effect; (b) a unified strength theory does not exist, and it was impossible to achieve a law of strength for various materials [20]. However, Yu first challenged and broke the second conclusion by his unified strength theory and twin-shear yield criterion. More importantly, Yu’s unified strength theory foreshadowed the existence of the strength law now that a unified formula of strength could be established [21]. Recently, Zhou established a structural state-of-stress theory and found that the essential strength of a material, regardless of the specimen’s size effect, exists in the uniaxial testing process of concrete specimens [4]. He indicated that the existing material strength defined at the specimen’s ultimate loads confused material strength and specimen’s strength, which surely led to the uncertainty of material strength. He defined the strength of concrete according to a natural law rather than the existing empirical and statistical bases, resulting in the discovery of the essential strength for concrete materials basically without the specimen’s size effect. Furthermore, Zhou determined the formula of strength (the law of strength) through biaxial and triaxial tests of concrete specimens under the assumption of homogeneous and isotropic materials. Both essential strength and formula of strength broke the two common conclusions mentioned above, which explored the new way to cast off the puzzling of the inconsistence and uncertainty of membrane material strengths. Hence, this study applied structural state-of-stress theory and methods to investigate the strain and stress data recorded in the uniaxial and biaxial tests of ETFE membrane material specimens. From the uniaxial and biaxial state-of-stress evolution curves, the characteristic points were detected via the cluster analysis method. For the uniaxial case, the characteristic points revealed the essential strength and essential design strength of ETFE membrane materials. For the biaxial case, the characteristic points derived essential principal stresses. Both essential principal stresses and essential strength bring about the unified formula of strength and derive the unified strength condition, which provide a new basis to update existing design codes for ETFE membrane materials and structures.

## 2. Essential Strength and Formula of Strength

Zhou et al. [4] initially investigated the strength of a material based on the mutation features in the state-of-stress evolutions of concrete specimens (uniaxial, biaxial, and triaxial tests) according to the natural law from quantitative change to qualitative change of a system, rather than from the specimen’s ultimate load-bearing state or specific strain value. This resulted in the discovery of the essential material strength (Ω) basically without the specimen’s size effect (uniaxial specimens). Furthermore, based on the mutation feature in the state-of-stress evolutions (biaxial and triaxial specimens), the author found a unified formula of strength for homogeneous and isotropic materials:(5)$$\Omega_{1}-\frac{1}{4}\Omega_{2}-\frac{3}{4}\Omega_{3}=\Omega$$
where Ω is the essential material strength, and $\Omega_{1}$, $\Omega_{2}$ and $\Omega_{3}$ are the combined principal stresses. For Equation (5), the principal stress $\Omega_{1}$ and the uniaxial strength Ω have the same sign expressing tension (+) or compression (−) and also define the order of triaxial principal stresses ($\Omega_{1}$, $\Omega_{2}$, $\Omega_{3}$).

The formula of strength represents a specific embodiment of the natural law from quantitative change to qualitative change of a system. Additionally, Equation (5) achieves the goal of a unified strength theory [21], offering a definite formula of strength for various materials. Hence, Equation (5) can be considered the law of strength for homogeneous and isotropic materials. Zhou’s discoveries directly provided a new physical law and the available methods to solve the classic and inherent problems of material strength. Further, according to Equation (5), Zhou derived the definite relationship between the essential shear strength $\tau_{\Omega }$ and the essential strength Ω:(6)$$\tau_{\Omega }=\frac{4}{7}\Omega ,$$
and the unified strength condition is
(7)$$\Omega_{1}-\frac{1}{4}\Omega_{2}-\frac{3}{4}\Omega_{3}\leq \left[\Omega \right]$$
where [Ω] is the allowable material strength referring to the essential strength Ω.

For the allowable material strength [Ω], Zhou discovered the principle of design strength embodied by the elastoplastic branch (EPB) feature in the state-of-stress evolution of uniaxial concrete specimens [22]. The strength defined at the EPB point of uniaxial concrete specimens could directly be taken as the design strength of concrete, i.e., the allowable material strength of concrete. Thus, the essential material strength, the EPB design strength, and the formula of strength collectively laid a new foundation for material strength that could update existing strength indexes, and design strength values and strength conditions of materials. The following studies on the ETFE membrane material strength provide the essential material strength, the EPB design strength, and a unified formula of strength.

## 3. Technical Route and Methods Adopted in this Study 

In this study, the uniaxial and biaxial tests of ETFE membrane specimens obtained the stress and strain data with an increase in tensile load. Then, state-of-stress analyzing methods were adopted to model and analyze the tested strain data of uniaxial and biaxial ETFE specimens. Finally, the essential strength and the unified formula of ETFE membrane material were set up according to the characteristic points in the state-of-stress evolution curves detected by the criterion.
Model the stress and strain data as generalized strain energy density (GSED) values using Equation (8) to express the state-of-stress of ETFE membrane specimens
(8)$$E_{ij}=\int_{0}^{\varepsilon_{ij}}\sigma d\varepsilon \;\mathrm{or}\;E_{ij}=\int_{0}^{\sigma_{ij}}\varepsilon d\sigma$$
where $E_{ij}$ and $F_{ij}$ are the GSED and load values at the *i*th tested point and the *j*th loading step; $\sigma_{ij}$ and $\varepsilon_{ij}$ are the stress and strain at the *i*th tested point and the *j*th loading step. Take $E_{ij}$ to characterize the state-of-stress of uniaxial specimen, called as the state-of-stress characteristic parameter. For a biaxial specimen, the sum of the GSED values along two directions is taken as the state-of-stress characteristic parameter. Then, plot the $E_{ij}-F_{ij}$ or $E_{ij}-\sigma_{ij}$ curve to illustrate the state-of-stress evolution. Apply the cluster analysis (CA) criterion to detect the characteristic points in the state-of-stress evolution curve, the $E_{ij}-F_{ij}$ curve, or the $E_{ij}-\sigma_{ij}$ curve [23]. The operative steps of the CA criterion is:Suppose that the possible characteristic point is *τ* and calculate the sum of the squares of deviations before and after *τ*:(9)$$V_{\tau }=\overset{\tau }{\underset{I=1}{\sum }}{\left(E_{i}-{\bar{E}}_{\tau }\right)}^{2},V_{n-\tau }=\overset{\tau }{\underset{I=\tau +1}{\sum }}{\left(E_{i}-{\bar{E}}_{n-\tau }\right)}^{2}$$Calculate the sum of $V_{\tau }$ and $V_{n-\tau }$:(10)$$S_{n}\left(\tau \right)=V_{\tau }+V_{n-\tau }$$Find the optimal bisection point $\tau_{m}$ using the criterion:(11)$$S\left(\tau_{m}\right)=\min\left\{S_{n}\left(\tau \right)\right\}\left(2\leq \tau \leq n-1\right)$$Point $\tau_{m}$ can then be inferred as the most possible characteristic point in the GSED sum-stress curve.Define the essential strength of ETFE membrane materials according to the state-of-stress characteristic point ($\tau_{m}$) of the uniaxial ETFE membrane specimen. Define the essential design strength of ETFE membrane materials according to the state-of-stress characteristic point ($\tau_{\mathrm{EPB}}$) of the uniaxial ETFE membrane specimen. Extract the combined principal stresses from the biaxial specimen, i.e., obtain the essential principal stresses at the state-of-stress characteristic point ($\tau_{m}$) in the $E_{ij}-F_{ij}$ or $E_{ij}-\sigma_{ij}$ curve. Find the relationship between the essential strength and essential principal stresses based on the formula of strength given in Equation (5), i.e., derive the unified formula of strength for ETFE membrane materials.Establish the unified strength condition for ETFE membrane materials according to Steps 5 and 6 above.

## 4. Essential Strength and Essential Design Strength

For ETFE membrane materials (AGC, Tokio, Japan), standard specimens commonly adopt rectangular shapes, with uniaxial tests used to derive the material strength, as shown in Figure 2. The size of the specimen refers to the existing standard and experimental apparatus [17,24]. The loading process is carried out at different strain rates because the ETFE membrane material relates to the loading speed. As the strain rate is consistent with the extension speed of the specimen, so this study treats the strain rate as approximately equal to the extension speed. The test records the displacements and acting tensile forces at the clamped edges of the specimen, which can obtain the corresponding strains and stresses used below. 

Figure 3 presents the E−σ curves of five rectangular specimens under a 100%/min loading rate. By applying the CA criterion, two characteristic points can be detected in the E−σ curves. According to structural state-of-stress theory [4,23], the characteristic points define the 1st and 2nd essential strengths (Ω, Ω′), as listed in Table 1 and Table 2. The essential strengths are not absolutely accurate, but they approach the essence of material strength. Table 1 and Table 2 also list the variances of Ω and Ω′ under various strain rates. It can be seen that the Ω values under individual strain rates are stable and accurate, indicating that Ω should be defined according to the characteristic point in the state-of-stress evolution of the uniaxial specimen. When compared with the Ω values, the Ω′ values have a slightly greater variance, suggesting that the initial random defects of specimens may begin to have an impact after Ω. 

For the E−σ curves in Figure 3, the CA criterion can also detect the other characteristic points, called the elastoplastic branch (EPB) points [4]. The EPB point can derive the strength directly as the design strength because it meets the basic requirement of an engineering design, as shown in Figure 3. Hence, the essential design strength of ETFE membrane materials is defined according to the specimen’s EPB feature, as listed in Table 3. The essential design strength of the ETFE membrane material is unique and definite, but the existing design strength is empirical and statistical since it adopts the coefficients of 3.5~7 [17]. In addition, the essential elastic modulus can be determined by the EPB point, as listed in Table 4.

## 5. Comparison of ETFE Material Strengths

Previously, there were not the officially issuing design standards for ETFE membrane structures. Engineering enterprises instead referred to the method for fabricating membrane materials alongside their own experience, and used the reduction factor method to suggest the tensile strength for design. The partial coefficient of membrane material resistance ($\gamma_{R}$) in the early code [25] was 3.5 for a short-term load and 7 for a long-term load. The tensile strengths with different extension speeds are further shown in Table 5. For the ETFE membrane parts in general positions, the strength reduction factor (ζ) was 1.0. The standard value of tensile strength ($f_{K}$) was the ultimate strength (σ) in the early code. Using Equation (3), the tensile strength design values in the early code ($f_{\mathrm{ea}}$) including short ($f_{\mathrm{ea-s}}$) and long ($f_{\mathrm{ea-1}}$) terms could be expressed by Equation (12) and are shown in Table 5.
(12)$$f_{\text{ea-s}}=\frac{f_{K}}{\gamma_{\text{R-s}}}=\frac{\sigma }{3.5},\;f_{\text{ea-l}}=\frac{f_{K}}{\gamma_{\text{R-l}}}=\frac{\sigma }{7}$$

Recently, researchers [11,26,27] found that the strain to the 2nd yield strength of ETFE membrane materials was about 15% to 16%. After the 2nd yield point was exceeded, the strain of the ETFE membrane material increased rapidly and smoothly until the breaking elongation of 300%. Therefore, it was unreasonable to use the breaking tensile strength as the standard tensile strength value, and difficult to determine the reduction factor as well. Figure 4a shows the σ−ε curves of five rectangular specimens under a strain rate of 100%/min. Figure 4b shows the strength points defined by the existing design code [17] (the geometrical and mathematical method), i.e., Points B and C are the 1st and 2nd yield stresses, respectively.

For the ETFE membrane material, the partial coefficient ($\gamma_{R}$)in the existing design code [17] is 1.2 for a short-term load and 1.8 (non-air support) or 1.4 (air support) for a long-term load. The strength reduction factor (ζ) is 1.0. The standard value of tensile strength ($f_{K}$) is the 1st yield stress ($\sigma_{S}$) in non-air support and the 2nd yield stress (${\sigma^{\prime }}_{S}$) for air-support. Using Equation (3), the tensile strength design value in the existing design code ($f_{\mathrm{ex}}$) including short ($f_{\text{ex-s}}$) and long ($f_{\text{ex-l}}$) terms can be expressed by Equation (13), as shown in Table 6.
(13)$$f_{\text{ex-s}}=\frac{f_{\text{K-s}}}{\gamma_{\text{R-s}}}=\frac{{\sigma^{\prime }}_{s}}{1.2},\;f_{\text{ex-l}}=\frac{f_{\text{K-l}}}{\gamma_{\text{R-l}}}=\frac{\sigma_{s}}{1.8}$$
where $\gamma_{\text{R-s}}$ and $\gamma_{\text{R-l}}$ is the partial coefficient of membrane material resistance for a short-term load and a long-term load.

Table 6 lists the 1st and 2nd essential strengths under different strain rates derived by detecting the characteristic points in the E−σ curves of rectangular specimens. Additionally, Figure 5 illustrates the distributions of the essential strength (Ω), the yield stress ($\sigma_{S}$) and the ultimate strength (σ) under different strain rates. It can be seen that the essential strengths at individual strain rates maintain stable and accurate values, indicating that the essential material strength has definite properties. 

The essential strength (Ω) and EPB strength ($\Omega_{\mathrm{EPB}}$) can be applied to establish the design strength of FTFE membrane structures as follows:Referring to the existing design code [17], the partial coefficient of membrane material resistance ($\gamma_{R}$) is 1.2 for a short-term load. Thus, the essential strength (Ω) is taken as the standard value of tensile strength ($f_{K}$). The design value of tensile strength ($f_{\mathrm{es}-s}$) for a short-term load can be expressed as Equation (14):(14)$$f_{\mathrm{es}-s}=\frac{f_{K}}{\gamma_{R-s}}=\frac{\Omega }{1.2}$$The essential EPB strength ($\Omega_{\mathrm{EPB}}$) is directly used as the design value of tensile strength ($f_{\mathrm{es}-l}$) for a long-term load, as expressed by Equation (15):(15)$$f_{\mathrm{es}-l}=\Omega_{\mathrm{EPB}}$$

Figure 6 shows the values of $f_{\mathrm{ea}-s}$,$f_{\mathrm{ea}-l}$,$f_{\mathrm{ex}-s}$,$f_{\mathrm{ex}-l}$,$f_{\mathrm{es}-s}$ and $f_{\mathrm{es}-l}$ derived from five rectangular specimens under different strain rates. It can be seen that the essential design strengths ($f_{\mathrm{es}-s}$,$f_{\mathrm{es}-l}$) are more stable and accurate, indicating their attribute of certainty.

## 6. Unified Strength Condition of ETFE Membrane Material

### 6.1. Formula of Strength for ETFE Membrane Materiasl

ETFE membrane materials are generally applied under a biaxial tensile state in engineering. From Equation (5), the unified formula of strength for biaxial tensile principal stresses can be written as
(16)$$\Omega_{1}-\frac{1}{4}\Omega_{2}=\Omega$$
where Ω is the essential strength, and $\Omega_{1}$ and $\Omega_{2}$ are the essential principal stresses. 

Next, the essential principal stresses from tests of the air bubbling models and the cruciform specimens made from the ETFE membrane material can be obtained [5,28]. Additionally, the stress and strain data are modelled as the GSED values using Equation (8), and their sum is taken as the state-of-stress characteristic parameter. Then, the CA criterion can detect the characteristic points in the E−σ curves. In this way, the biaxial tensile stresses at the characteristic points are taken as the essential principal stresses ($\Omega_{1}$, $\Omega_{2}$) corresponding to the first essential strength (Ω). Ω and ($\Omega_{1}$, $\Omega_{2}$) are then derived at the characteristic points detected by the CA criterion in the state-of-stress evolution curves of uniaxial and biaxial specimens.

The principal stresses at the top point of the circular ETFE membrane bubbling model can be calculated as [28]
(17)$$\Omega_{1}=\Omega_{2}=\frac{pR}{2t_{p}}$$
where *t_p_* is the thickness of the ETFE membrane material model at the pole; *p* is air pressure; *R* is the radius of the model. *t_p_* can be calculated as
(18)$$t_{p}=\frac{t_{0}}{{\left(1+\frac{h^{2}}{a^{2}}\right)}^{2}}$$
where $t_{0}$ is the initial thickness of the ETFE membrane material model, and a is the radius of the model before its deformation. Figure 7 shows the E−σ curves and the principal stresses ($\Omega_{1}$,$\;\Omega_{2}$) defined by the CA criterion.

Figure 8 and Figure 9 show the characteristics points in the E−σ curves of cruciform specimens and the characteristic points detected by the CA criterion, as well as the corresponding essential principal stresses ($\Omega_{1}$, $\Omega_{2}$). The slight difference in the stresses ($\Omega_{1}$, $\Omega_{2}$) could be due to the errors of the measurements in two directions.

After obtaining the essential strength and the essential principal stresses, the validity of Equation (16) can be verified by estimating the errors between the equivalent stress $\Omega_{\mathrm{eq}}$ = $\Omega_{1}-\frac{1}{4}\Omega_{2}$ with the first essential strength Ω or the equivalent stress $\Omega \prime_{\mathrm{eq}}$ = $\Omega \prime_{1}-\frac{1}{4}\Omega \prime_{2}$ with the 2nd essential strength Ω′. Here, the percentage η is set to estimate the error:(19)$$\eta =\frac{\left|\Omega_{\mathrm{eq}}-\Omega \right|}{\Omega }\%\;\mathrm{or}\;\eta^{\prime }=\frac{\left|{\Omega^{\prime }}_{\mathrm{eq}}-\Omega^{\prime }\right|}{\Omega^{\prime }}\%$$

Table 7 lists the errors (η) of Equation (16) verified by the tests of bubbling models and cruciform specimens. It can be seen that the essential principal stresses ($\Omega_{1}$, $\Omega_{2}$) and the first essential strength (Ω) accurately fit the formula of strength in Equation (16). It needs to indicate that Equation (16) fits the essential principal stresses ($\Omega_{1}$, $\Omega_{2}$) from the cruciform specimens or the air bubbling models to the first essential strength (Ω) under 25%/min or 500%/min, respectively.

Similarly, the essential principal stresses ($\Omega \prime_{1}$, $\Omega \prime_{2}$) corresponding to the 2nd essential strength (Ω′) can be derived by detecting the characteristic points in the E−σ curves with the CA criterion. Table 8 lists the errors (η′) of Equation (16) verified by the tests of the bubbling models and cruciform specimens. It can be seen that ($\Omega \prime_{1}$, $\Omega \prime_{2}$) and Ω′ accurately fit the formula of strength in Equation (16).

### 6.2. Advanced Design of ETFE Membrane Structures

The essential strength reflects the essence of a material strength, i.e., a material strength should be unique and unaffected by the specimen’s size and shape effects. Comparably, the existing strength of a material is derived from the ultimate loads of specimens with considerable variation and related with specimen’s shape and size effects. The 1st yield strength derived by Wu’s method represents progress for ETFE membrane materials because it utilizes the trend change in the stress–strain curves of specimens. However, the 1st yield strength was not applied to derive the design strength, only as a safe estimation to the design strength referring for the ultimate strength. 

Corresponding to Equation (7), the unified strength condition of ETFE membrane materials can be set as
(20)$$\Omega_{1}-\frac{1}{4}\Omega_{2}\leq \Omega_{\mathrm{EPB}}$$
where $\Omega_{\mathrm{EPB}}$ is the EPB strength that replaces the allowable stress ([Ω]).

The traditional researches and this study fundamentally seek to reach the same goal in order to accurately define the strength and determine a formula for strength. At present, the strength of ETFE membrane materials is commonly defined by the tensile strength derived from empirical and statistical judgment. This judgment is difficult to derive the accurate strength index and the unified formula of strength. The difficulty could be overcome though state-of-stress analysis of strain data obtained from tests of uniaxial and biaxial ETFE membrane material specimens.

## 7. Conclusions

Structural state-of-stress theory led to discoveries of the essential strength and the formula of strength for homogeneous and isotropic materials. Based on this foundation, this study conducts the state-of-stress modeling and analysis of the tested stress–strain data of ETFE membrane material specimens. The achieved results can draw the following conclusions. 

The essential strength of ETFE membrane material should be defined at the characteristic points in the state-of-stress evolution curves of uniaxial specimens, which has the attribute basically without specimen’s size effect. Additionally, the essential design strength of ETFE membrane material should be defined at the EPB points in the state-of-stress evolution curves of uniaxial specimens. EPB design strength has the stable, accurate and unique features, with two margins of safety: One is from the EPB strength point to the essential strength point with an attribute of certainty and the other is from the essential strength point to the ultimate strength point with an attribute of semi-certainty. Hence, the EPB strength could bring the accurate design as well as the accurate estimation of safety in the engineering applications of ETFE membrane materials.

The formula of strength for homogeneous and isotropic materials (Equation (5)) yields a unified formula of strength for ETFE membrane materials expressed by Equation (16). Furthermore, the unified strength condition of ETFE membrane materials can be derived as Equation (20), which establishes the definite relationship between the essential design strength and the essential principal stresses. Thus, this study could explore a new way to address the classic issues of inconsistent strength indexes and empirical strength conditions of ETFE membrane materials.

## Figures and Tables

**Figure 1 polymers-14-05166-f001:**
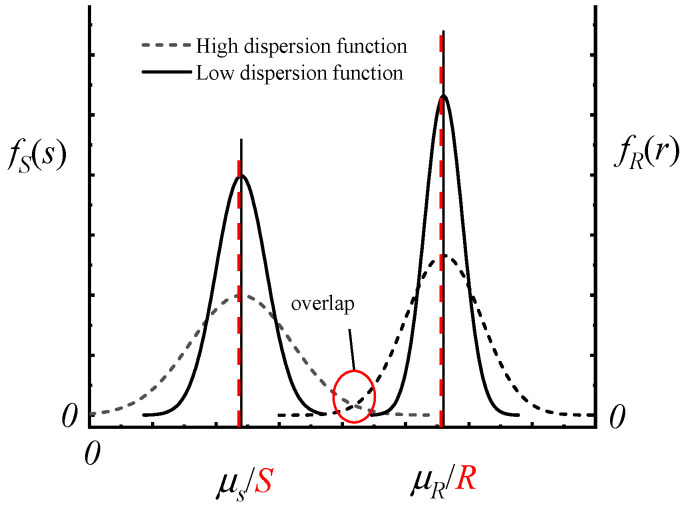
The probability density functions $f_{R}(r)$ and $f_{S}(s)$ of *R* and *S*.

**Figure 2 polymers-14-05166-f002:**
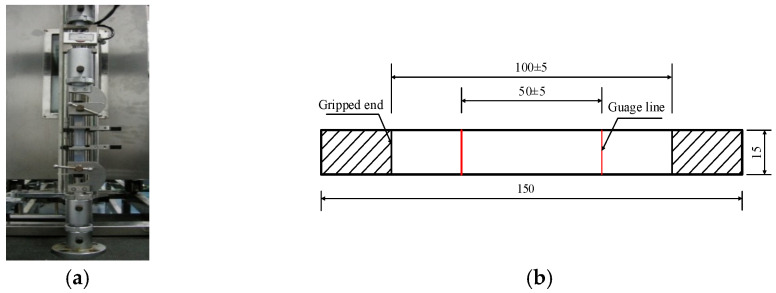
The experimental apparatus (NSS, Shenzhen, Guangdong, China) and ETFE membrane specimen. (**a**) Experimental apparatus. (**b**) Rectangular specimen (mm).

**Figure 3 polymers-14-05166-f003:**
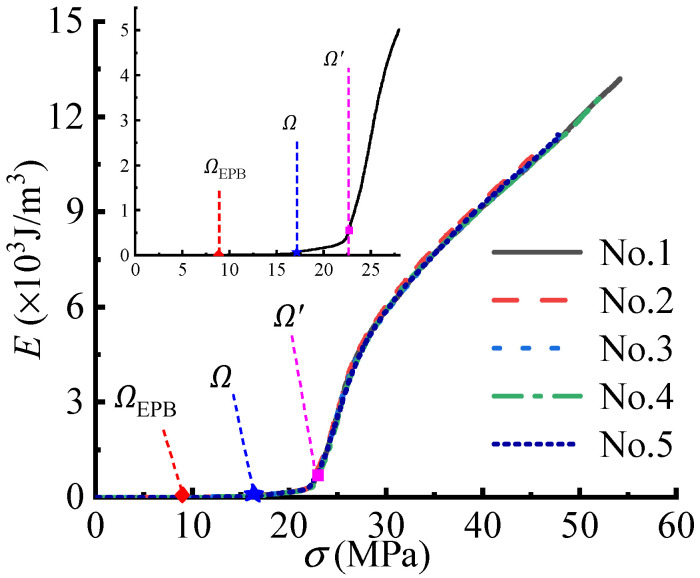
The E−σ curves and the defined essential strengths (Ω, Ω′).

**Figure 4 polymers-14-05166-f004:**
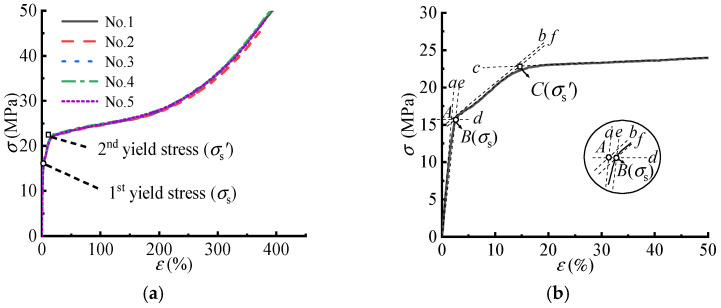
The σ−ε curves and strengths defined by the existing design code. (**a**) The σ−ε curves. (**b**) The points defining the yield stresses.

**Figure 5 polymers-14-05166-f005:**
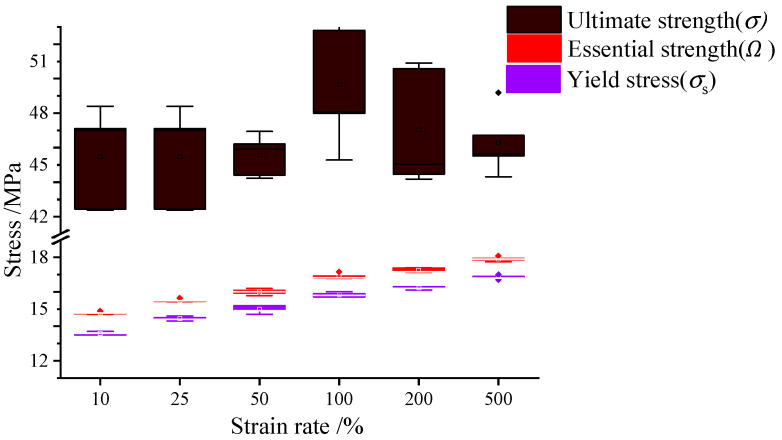
The distributions of essential strengths, yield stresses, and ultimate strengths.

**Figure 6 polymers-14-05166-f006:**
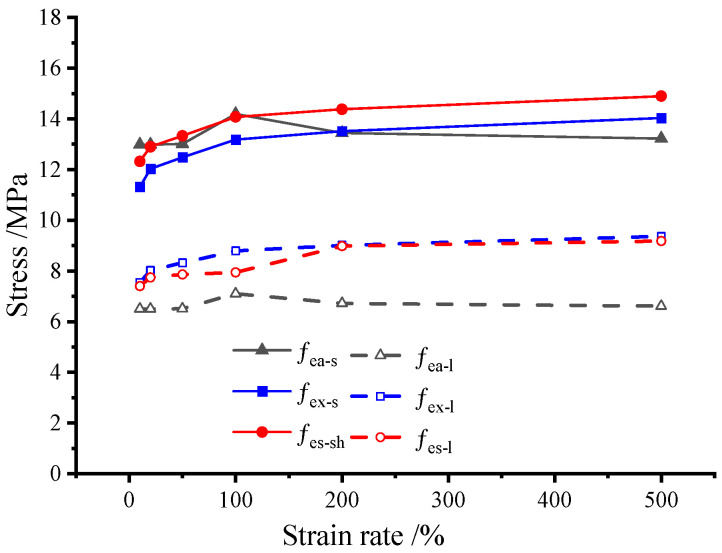
The comparison of individual design strengths.

**Figure 7 polymers-14-05166-f007:**
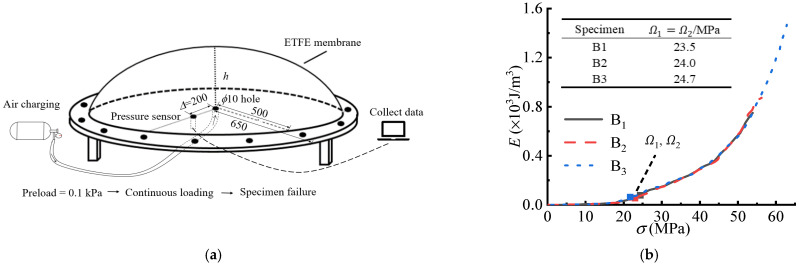
The air bubbling model and detected essential principal stresses ($\Omega_{1}$,$\;\Omega_{2}$). (**a**) The air bubbling model. (**b**) The E−σ curves and the essential principal stresses ($\Omega_{1}$,$\;\Omega_{2}$).

**Figure 8 polymers-14-05166-f008:**
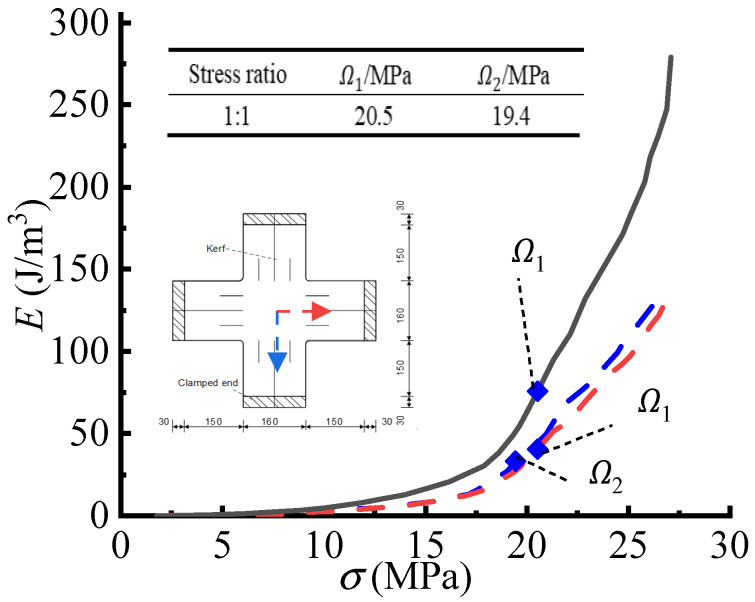
The E−σ curves of the cruciform specimen (1:1) and the detected essential principal stresses.

**Figure 9 polymers-14-05166-f009:**
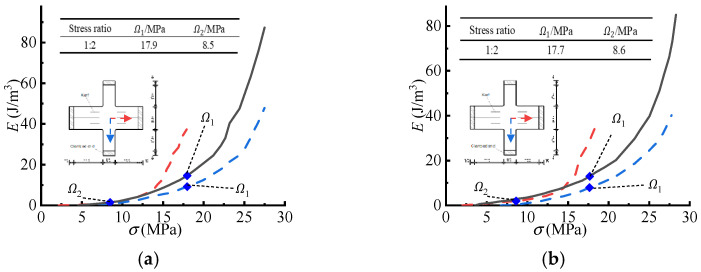
The E−σ curves of the cruciform specimens (1:2) and the detected essential principal stresses: (**a**) specimen 1, (**b**) specimen 2.

**Table 1 polymers-14-05166-t001:** The 1st essential strength values (Ω) of ETFE membrane material under different strain rates.

Strain Rate	The 1st Essential Strengths Ω/Mpa					Average $\bar{\Omega }$/MPa	Variance
	No.1	No.2	No.3	No.4	No.5		
10%/min	14.7	14.7	14.8	14.8	14.9	14.8	0.005
25%/min	15.4	15.4	15.6	15.5	15.5	15.5	0.006
50%/min	16.2	16.0	15.8	16.1	15.9	16.0	0.025
100%/min	17.2	16.8	16.9	16.8	16.8	16.9	0.023
200%/min	17.1	17.2	17.4	17.2	17.4	17.3	0.018
500%/min	18.1	17.9	17.8	17.7	17.9	17.9	0.015

**Table 2 polymers-14-05166-t002:** The 2nd essential strength values (Ω′) of ETFE membrane material under different strain rates.

Strain Rate	$\mathrm{The}\;2\mathrm{nd}\;\mathrm{Essential}\;\mathrm{Strength}\;\Omega^{\prime }/\mathrm{MPa}$					$\mathrm{Average}\;{\bar{\Omega }}^{\prime }/\mathrm{MPa}$	Variance
	No.1	No.2	No.3	No.4	No.5		
10%/min	22.2	22.2	22.2	21.7	22.5	22.2	0.083
25%/min	22.8	22.2	23.0	22.3	23.0	22.7	0.148
50%/min	23.2	22.9	23.0	23.1	22.0	22.8	0.233
100%/min	23.1	23.2	23.7	23.1	23.8	23.4	0.212
200%/min	24.4	23.7	24.3	23.3	23.7	23.9	0.117
500%/min	24.6	24.3	24.1	24.2	24.2	24.3	0.037

**Table 3 polymers-14-05166-t003:** The essential design strength values ($\Omega_{EPB}$) of ETFE membrane material under different strain rates.

Strain Rate	The Essential Design Strength/MPa					$\mathrm{Average}\;{\bar{\Omega }}_{\mathrm{EPB}}/\mathrm{MPa}$	Variance	${\bar{\Omega }}_{\mathrm{EPB}}/\bar{\Omega }$
	No.1	No.2	No.3	No.4	No.5			
10%	8.1	6.8	6.8	6.7	8.6	7.40	0.6280	0.500
25%	7.8	7.7	7.8	7.5	7.9	7.74	0.0184	0.499
50%	8.9	8.0	6.8	7.2	8.4	7.86	0.5904	0.491
100%	7.9	7.9	8.1	7.8	8.0	7.94	0.0104	0.470
200%	9.5	9.3	8.6	9.1	8.4	8.98	0.1736	0.519
500%	8.8	8.9	10.5	8.9	8.8	9.18	0.4376	0.513

**Table 4 polymers-14-05166-t004:** The essential elastic modulus values of ETFE membrane material under different strain rates.

Strain Rate	The Essential Elastic Modulus/MPa					Average/MPa	Variance
	No.1	No.2	No.3	No.4	No.5		
10%	8.4	8.5	8.2	8.2	8.0	8.26	0.0304
25%	8.0	7.8	8.0	7.9	8.0	7.94	0.0064
50%	8.3	7.4	8.5	8.4	8.2	8.16	0.1544
100%	8.1	8.0	8.1	8.1	8.1	8.08	0.0016
200%	8.0	7.8	8.0	7.7	7.8	7.86	0.0144
500%	7.5	7.5	7.3	7.5	7.4	7.44	0.0064

**Table 5 polymers-14-05166-t005:** The tensile strength values of ETFE membrane material under different strain rates.

Strain Rate	The Tensile Strength σ/MPa					Average $\bar{\sigma }$/MPa	Variance	*f*_ea_/MPa	
	No.1	No.2	No.3	No.4	No.5			*f* _ea-s_	*f* _ea-l_
10%	43.8	45.1	42.0	44.2	44.7	43.96	1.15	12.56	6.28
25%	48.4	42.4	47.1	42.4	47.0	45.46	6.49	12.99	6.49
50%	44.2	46.9	45.9	44.4	46.2	45.52	1.10	13.01	6.50
100%	52.8	48.1	54.2	45.3	48.0	49.68	10.93	14.19	7.10
200%	45.0	44.2	50.6	44.5	50.9	47.04	9.25	13.44	6.72
500%	49.2	46.7	44.3	45.5	45.6	46.26	2.74	13.22	6.61

**Table 6 polymers-14-05166-t006:** The 1st and 2nd yield stresses ($\sigma_{s}\;,\sigma \prime_{s}$) determined by the existing design code.

Strain Rate		The 1st and 2nd Yield Stresses $(\sigma_{s}\;,\sigma_{s}^{\prime })\;/\mathrm{MPa}$					Average $\bar{\sigma }$/MPa	Variance	*f*_ex_/MPa	
		No.1	No.2	No.3	No.4	No.5			*f* _ex-s_	*f* _ex-l_
10%	$\sigma_{s}$	13.6	13.7	13.5	13.6	13.5	13.58	0.0056	7.54	11.32
	$\sigma \prime_{s}$	21.2	21.5	21.3	21	21.1	21.22	0.0296	15.16	17.68
20%	$\sigma_{s}$	14.4	14.5	14.6	14.4	14.3	14.44	0.0104	8.02	12.03
	$\sigma \prime_{s}$	21.6	21.5	21.8	21.6	21.6	21.62	0.0096	15.44	18.02
50%	$\sigma_{s}$	15.2	15.2	14.9	14.9	14.7	14.98	0.0376	8.32	12.48
	$\sigma \prime_{s}$	21.9	21.8	21.6	21.7	21.4	21.68	0.0296	15.46	18.07
100%	$\sigma_{s}$	16.0	15.7	15.7	15.9	15.8	15.82	0.0136	8.79	13.18
	$\sigma \prime_{s}$	22.5	22.1	22.2	22.3	22.2	22.26	0.0184	15.90	18.55
200%	$\sigma_{s}$	16.1	16.3	16.2	16.3	16.2	16.22	0.0056	9.01	13.52
	$\sigma \prime_{s}$	22.0	22.3	22.1	22.3	22.1	22.16	0.0144	15.83	18.47
500%	$\sigma_{s}$	17.0	16.8	16.7	16.9	16.8	16.84	0.0104	9.36	14.03
	$\sigma \prime_{s}$	22.7	22.4	22.5	22.5	22.5	22.52	0.0096	16.09	18.77

**Table 7 polymers-14-05166-t007:** The verification of Equation (16) using the data ($\Omega_{1}$, $\Omega_{2}$, and $\Omega_{\mathrm{eq}}$).

Biaxial Test	No. of Specimen		$\Omega_{1}/\mathrm{MPa}$	$\Omega_{2}/\mathrm{MPa}$	$\Omega_{\mathrm{eq}}/\mathrm{MPa}$	η/%
Air bubbling models	B1		23.5	23.5	17.6	1.7
	B2		24.0	24.0	18.0	0.6
	B3		24.7	24.7	18.5	3.3
Cruciform specimens	1:1		20.5	19.4	15.6	0.65
	1:2	1	17.9	8.5	15.8	1.94
		2	17.7	8.6	15.6	0.65

**Table 8 polymers-14-05166-t008:** The verification of Equation (16) using the data ($\Omega \prime_{1}$, $\Omega \prime_{2}$ and $\Omega \prime_{\mathrm{eq}}$).

Biaxial Test	No. of Specimen		$\Omega_{1}^{\prime }/\mathrm{MPa}$	$\Omega_{2}^{\prime }/\mathrm{MPa}$	$\Omega_{\mathrm{eq}}^{\prime }/\mathrm{MPa}$	$\eta^{\prime }/\%$
Air bubbling models	B1		34.3	34.3	25.7	5.8
	B2		34.1	34.1	25.6	5.3
	B3		35.4	35.4	26.6	9.5
Cruciform specimens	1:1		27.1	26.5	20.5	9.7
	1:2	1	27.5	13.9	24.0	5.7
		2	27.8	14.4	24.2	6.6

## Data Availability

The data presented in this study are available on request from the corresponding author.

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
