# Peer review of "Essential Design Strength and Unified Strength Condition of ETFE Membrane Material"

_polymers, 2022, doi:10.3390/polym14235166_

Round 1

Reviewer 1 Report

The manuscript "Essential design strength and unified strength condition of ETFE membrane material" by Zhang et al. describes the authors approach at unifying otherwise inconsistent strength indices for ETFE membranes. While I think the overall study has merits as to simplify the evaluation and design principles for ETFE membrane structures safety, there are some issues that needed to be adressed in the presentation.

First, I think extensive language editing is needed (in particular the use of articles a/the, "researches", odd phrasings).

In Figure 1: the labelling ("discrete"/"connected") seems mixed up.

In the main text, when describing the figure, maybe use "overlap" instead of "interfere" when discussing the curves.

`

Also there, the authors write: "The change process from the solid line to the dotted line implied that when the discreteness of resistance and action effect increased, the structural reliability would decrease but the safety factor did not change [11]."

This seems an unnecessary convoluted sentence (also can act as example why I suggest a general proof reading/language editing). But more important: wouldn't the structural reliability increase, too, whith increasing discretness?

The issues in language also made it hard to go through the other parts of the manuscript, though I did not find further direct errors in the later parts.

Reviewer 2 Report

In this manuscript, the authors proposed the essential design strength and the unified strength condition of ETFE membrane material. They modeled the characterization of the specimen’s state-of-stress on the tested strain data of uniaxial rectangle-shaped specimens. They used the cluster analysis (CA) criterion to detect the characteristic points in the state-of-stress evolution curve. The state-of-stress modeling and evolution analysis derived the essential principal stresses at the characteristic points detected by the CA criterion. The essential principal stresses satisfy the formula of strength with the essential strength, leading to the unified strength condition of ETFE membrane material. Finally, the essential strength, essential design strength, and the unified strength condition are compared with the existing ones, presenting the rationality to update the existing analysis and design methods for determining the strength of ETFE membrane material. This study is good and important to study the stree-stran correlation on the membranes. The interpretations of the results are well discussed. The quantity and quality of the figures are appropriate. However, recent references should be included in the text. We believe that this research subject is promising for studying the mechanical properties of membranes.  

Summary: I recommend publishing this manuscript after considering my comments on the attached file.

Round 2

Reviewer 1 Report

The revised manuscript has clearly benefited from the proof reading, however, I still have the same issue with the Figure 1 (Maybe the authors misunderstood my comment): Still, the overlapping functions (dashed lines) are labelled "discrete", while the non-overlapping (solid lines) are labelled "connected". This is a total misnomer, as "discrete" implies "seperate, discontinuous, disjoined...", while exactly these functions (the dashed lines) ARE connected over the overlap, while the others (solid lines) are not. Thus these labels are still mixed up.

A still remaining odd choice of words is when the authors refer to the "mutation points" in different curves. I guess the authors mean characteristics, critical, or transition points, or something in this line?
